# Optical Fiber-Based Recording of Climbing Fiber Ca^2+^ Signals in Freely Behaving Mice

**DOI:** 10.3390/biology11060907

**Published:** 2022-06-13

**Authors:** Jiechang Tang, Rou Xue, Yan Wang, Min Li, Hongbo Jia, Janelle M. P. Pakan, Longhui Li, Xiaowei Chen, Xingyi Li

**Affiliations:** 1Center for Neurointelligence, School of Medicine, Chongqing University, Chongqing 400030, China; 18860874809@163.com (J.T.); xr33225914@163.com (R.X.); 2College of Bioengineering, Chongqing University, Chongqing 400044, China; 3Brain Research Instrument Innovation Center, Suzhou Institute of Biomedical Engineering and Technology, Chinese Academy of Sciences, Suzhou 215163, China; wangyan@sibet.ac.cn (Y.W.); limin@sibet.ac.cn (M.L.); hongbo.jia@lin-magdeburg.de (H.J.); 4Combinatorial NeuroImaging Core Facility, Leibniz Institute for Neurobiology, 39118 Magdeburg, Germany; 5Institute of Cognitive Neurology and Dementia Research, Otto-von-Guericke University, 39120 Magdeburg, Germany; janelle.pakan@med.ovgu.de; 6German Center for Neurodegenerative Diseases (DZNE), 39120 Magdeburg, Germany; 7Brain Research Center and State Key Laboratory of Trauma, Burns, and Combined Injury, Third Military Medical University, Chongqing 400038, China; 8Guangyang Bay Laboratory, Chongqing Institute for Brain and Intelligence, Chongqing 400064, China

**Keywords:** the olivocerebellar circuitry, climbing fibers, optical fiber photometry, lobule IV/V of the cerebellar vermis, open field

## Abstract

**Simple Summary:**

In this study, we established a robust and accessible method to chronically record calcium signals from climbing fiber (CF) projections to the cerebellar cortex in freely behaving mice. This technique was demonstrated with optical fiber photometry in lobule IV/V of the cerebellar vermis during open field exploration, where various movement-evoked CF Ca^2+^ signals were observed, and the onset of exploratory-like behaviors was highly synchronous with the recorded CF activity.

**Abstract:**

The olivocerebellar circuitry is important to convey both motor and non-motor information from the inferior olive (IO) to the cerebellar cortex. Several methods are currently established to observe the dynamics of the olivocerebellar circuitry, largely by recording the complex spike activity of cerebellar Purkinje cells; however, these techniques can be technically challenging to apply in vivo and are not always possible in freely behaving animals. Here, we developed a method for the direct, accessible, and robust recording of climbing fiber (CF) Ca^2+^ signals based on optical fiber photometry. We first verified the IO stereotactic coordinates and the organization of contralateral CF projections using tracing techniques and then injected Ca^2+^ indicators optimized for axonal labeling, followed by optical fiber-based recordings. We demonstrated this method by recording CF Ca^2+^ signals in lobule IV/V of the cerebellar vermis, comparing the resulting signals in freely moving mice. We found various movement-evoked CF Ca^2+^ signals, but the onset of exploratory-like behaviors, including rearing and tiptoe standing, was highly synchronous with recorded CF activity. Thus, we have successfully established a robust and accessible method to record the CF Ca^2+^ signals in freely behaving mice, which will extend the toolbox for studying cerebellar function and related disorders.

## 1. Introduction

The roles of the cerebellum in motor learning and the coordination of muscle movements are well established—so much so that cerebellar function is often simplified to the precise control of elementary movements. Underlying this generalization are the relatively simple and morphologically uniform cellular connections found throughout the cerebellum, which have been well known for over a century [1]. For instance, the olivocerebellar circuitry, consisting of climbing fiber (CF) input originating from inferior olivary (IO) neurons that provide afferent input to cerebellar Purkinje cells (PCs), has been mapped in precise detail [2], and is known to play an important role in motor coordination [3] and motor learning [4,5,6]. This CF input was traditionally described as inducing an all-or-none response in PCs [7,8] and formed an important basis as a “teaching” signal within theories of cerebellar learning [7]. However, the binary nature of CF signals has recently come into question as new theories [9], and experimental recordings in vivo under behaving conditions, have suggested that CFs may have graded responses represented by different levels of bursting, which may provide a more instructive signal for learning [10,11,12]. However, direct recording of the activity of CF terminals has not been widely applicable in freely behaving animals. To address these questions and update long-standing cerebellar learning algorithms, it will be important to establish robust methods for directly monitoring CF signals in freely behaving mice.

Several methods are currently established to observe the dynamics of the olivocerebellar circuitry [13,14]. For instance, in vitro recordings have provided fundamental information regarding the generation of simple spikes and complex spikes [15], the physiological features of dendrites [16], and the functional properties of both PC and IO neurons [13,17]; however, results can be difficult to translate to in vivo circuitry [10,18]. Electrical stimulation of the IO and electrophysiological recording of complex spike activity in PCs have been widely used to study the functional features of cerebellar processing in vivo [13,19,20]; however, these techniques are highly technically challenging to apply to CFs, especially in freely behaving animals. Furthermore, two-photon microscopy has been a useful tool to directly monitor the activity of IO neurons [21], CFs [22], and PCs [23] in vivo, and can be combined with other sophisticated techniques, such as electrophysiological recording [24,25] or optogenetics [26,27]. However, these techniques require expensive and complex imaging set-ups that limit accessibility and are also not compatible with freely moving behaviors, generally requiring head-fixed animals. Since the cerebellum plays a large role in various types of motor behaviors requiring balance, including extensive integration of vestibular input [28], refined methods for assessing CF activity in freely moving animals are vital. Therefore, in this study, we sought to establish an accessible, low-cost, robust method for recording CF signals in freely behaving mice.

To do this, we established a protocol for optical fiber photometry [29] combined with genetically encoded calcium indicators (GECIs) for the direct recording of CF signals in freely behaving mice. Recording calcium signals from CFs poses several challenges. First, because of the topographic organization of CF projections to the cerebellar cortex, it can be difficult to precisely target the injection site in the IO for maximal efficiency of CF labeling at the desired recording site in the cerebellar cortex. Second, the fixation of an optical fiber targeting the molecular layer is also difficult due to the proximity to the brain surface. Finally, the specific expression profile of the GECIs needs to be tested and the labeling density needs to be verified. Therefore, we developed a protocol to overcome these challenges and apply this methodology to examine precise motor behavior and CF signals in freely behaving mice in an open field paradigm. Thus, we established a highly accessible method to record the Ca^2+^ signals of CF in freely behaving mice, which may extend the toolbox for studying the fundamental parameters of cerebellar function and related cerebellar disorders.

## 2. Material and Methods

### 2.1. Animals

Male C57BL/6J mice (age 2–3 months, weight 22–25 g; SPF (Beijing) Biotechnology Co., Ltd.; *n* = 33) were used for the experiments. Mice were group-housed before implantation of optical fibers and separated into individual home cages that were compatible with the optical fiber implantation after surgery and for the remainder of the experiment. Animals were kept under a 12 h light/12 h dark cycle (lights on at 7:00 a.m.) with free access to food and water. All animal procedures were performed in accordance with the Experimental Animal Welfare Ethics Committee of Chongqing University (Approval Date 15 October 2021; Code No. CQU-IACUC-RE-202205-002).

### 2.2. Surgery

Mice were anesthetized with 2.5% isoflurane (mixed with air at a flow rate of 0.5 L/min; induction lasted for approximately 2 min, until the breathing rate slowed and there was an absence of the toe-pinch reflex) and fixed on a stereotaxic frame with a heating plate (maintained at 36.5–37.5 °C using a digital thermometer). General anesthesia was maintained with 1.0–1.2% isoflurane until the end of surgery, with the depth of anesthesia monitored by the respiratory rate and toe-pinch reflex. Both eyes were protected by ophthalmic ointment to prevent drying. To clean the surface of the skull after removing the hair and skin, 10% hydrogen peroxide (H_2_O_2_) and 0.1 M phosphate-buffered saline (PBS) were used. Lidocaine (2%) was applied to the surface of the skin before incision.

For anterograde tracing of CF terminals, injections into the IO were made. A small craniotomy (diameter ~0.5 mm) was performed above the cerebellar cortex with a dental drill at 2.85 ± 0.03 mm posterior and 0.40 ± 0.03 mm lateral from lambda. For retrograde tracing of the IO, a small craniotomy (diameter ~0.5 mm) was performed above the cerebellar vermis lobule IV/V with a dental drill at 2.40 ± 0.03 mm posterior from lambda and 0.45 ± 0.03 mm lateral from midline; a retrograde tracer (cholera-toxin B, CTB 555, Invitrogen; 50 nl) was pressure-injected with a glass micro-pipette. For experiments with the optical fiber recording of CF signals, a craniotomy (~1 mm) was performed above lobule IV/V at 2.40 mm posterior and −0.70 mm lateral to lambda.

### 2.3. Virus Injection

An AAV driving the expression of either enhanced green fluorescent protein (EGFP) or mCherry was injected for the anterograde tracing of CFs. In 6 mice, approximately 50 nL of AAV2/8-hSyn-EGFP-WPRE-pA was injected into the right IO and approximately 50 nL of AAV2/9-hSyn-mCherry-WPRE-pA was injected into the left IO, both at a depth of 5.50 ± 0.03 mm, using a glass micro-pipette with a tip diameter of 10–20 µm. 

In additional experimental mice, an AAV driving the expression of the GECI, AAV2/9-hSyn-axon-jGCaMP7b, was injected (100 nl) into the IO for the labeling of CFs and subsequent recording of calcium activity. Before injection, the pipette loaded with AAV was held in place for an additional 1 min. The injection rate was approximately 5 nL/s. After injection, the pipette was held in place for an additional 5 min before retraction. Then, the pipette tip was gradually lifted by 50 μm and held in place for 2 min. This procedure was repeated three times. After slowly retracting the pipette from the brain and closing the scalp with tissue glue (Vetbond, USA), the mice recovered in their home cage and Meloxicam (5 mg/kg; Metacam, Germany) was provided by subcutaneous injection. After 4 weeks to allow for AAV expression, the mice were used for either histological examination or optical fiber recording.

### 2.4. Optical Fiber Recording Set-Up

A custom-built optical fiber recording set-up (Dual-Channel Fiber Photometry, Thinker Tech Nanjing Biotech Co., Ltd., Nanjing, China) and software (model “Fiber Photometry v3.0,” Thinker Tech Nanjing Biotech Co., Ltd., Nanjing, China) were used to acquire and analyze Ca^2+^ transients resulting from the change in fluorescence (ΔF/F) of the GECI. The light intensity of LED excitation (wavelength 470 nm or 580 nm) through the optical fiber was approximately 0.375 mW/mm^2^ at the fiber tip (diameter 200 µm; Numerical Aperture (NA), 0.37; Inper, Hangzhou, Zhejiang, China). The excitation light passed through multiple dichroic mirrors (MD498; Thorlabs) and converged through the objective lens (NA = 0.4, Olympus). Then, the excitation light reached the sample through the optical fiber. Emission light generated by the calcium indicator was collected by the optical fiber and detected by a highly sensitive photomultiplier tube (PMT, H10721-210, Hamamastsu, Japan), and then the signals were output as voltage and were recorded in real time after filtering and amplification by a data acquisition card (DAQ, Thinker Tech Nanjing Biotech Co., Ltd., Nanjing, China).

### 2.5. Optical Fiber Recordings in Freely Behaving Mice

Following anesthesia, a craniotomy (~1 mm) was performed above lobule IV/V at 2.05 mm posterior and 0.70 mm lateral to lambda. A cannula (inner diameter 0.57 mm, outer diameter 0.81 mm) was glued to the skull above the craniotomy site and the tip of the optical fiber was extended 0.9 mm out of the cannula. The optical fiber was placed above the craniotomy site and advanced into the cannula up to 500 µm below the surface of the dura. The skull surface was roughened to improve adhesion, and the optical fiber and cannula were then fixed in situ using dental cement (PALA, Germany). After 10 min to solidify, a black pigment was applied to prevent light interference. 

Following 24 h recovery, each mouse was individually placed in a gray rectangular open field environment (40 × 40 × 40 cm); the optical fiber implanted in the mouse was connected to the recording set-up by an FC-type connector, and fluorescent signals were recorded using the optical fiber. In addition, two cameras (sampling rate = 30 Hz) were placed at opposite positions, covering the entire area of the box, to monitor the animal’s behavior. The behavior of the mouse and the Ca^2+^ transients of the labeled CFs were recorded simultaneously. Each animal was recorded for 20 min at a time, 3 times a day, for, at most, 7 days. Mice were returned to their home cages after each recording for at least 20 min. After daily recording sessions were complete, each mouse was returned to its custom home cage and the optical fiber was secured to prevent bending. In addition, the recording box was cleaned thoroughly with 70% ethanol. Control (EGFP) mice were recorded similarly.

### 2.6. Histology and Fluorescence Imaging

The mice were deeply anesthetized using pentobarbital (90 mg/kg) and perfused transcardially with 0.9% saline followed by 4% paraformaldehyde (PFA) solution. Brains were extracted and immersed in 4% PFA with 10% sucrose for cryoprotection overnight. Each sample was continuously sliced into 40 µm slices, and then stained with 4′,6-diamidino-2-phenylindole (DAPI). Images of serial sections were acquired using a fluorescence microscope (OLYMPUS, Tokyo, Japan). Selected slices were used for immunohistochemistry to confirm the distribution of CF in the cerebellar cortex. Briefly, slices were blocked at room temperature for 2 h in 10% donkey serum and 0.3% Triton X-100 in PBS and then incubated with the primary antibodies at 4 ℃ for 24 h. After washing, the sections were incubated with secondary antibodies at room temperature for 2 h in 5% donkey serum and 0.3% Triton X-100 in PBS. Here, two kinds of PC markers (PC protein 4 (PCP4) and Calbindin-D28k) were used. For staining with PCP4, the primary antibodies used were rabbit anti-PCP4 (1:200, Sigma, St. Louis, MI, USA) and chicken anti-GFP (1:500, Abcam, Cambridge, UK). The following secondary antibodies were used: Alexa Fluor 594-conjugated donkey anti-rabbit IgG (1:500, Invitrogen, Waltham, MA, USA) and Alexa Fluor 488-conjugated donkey anti-chicken IgG (1:1000, Sigma). For staining with Calbindin-D28K, the primary antibodies used were mouse anti-Calbindin-D28K (1:1000, Sigma) and chicken anti-GFP (1:500, Abcam), followed by secondary antibodies: Alexa Fluor 568-conjugated donkey anti-mouse IgG (1:1000, Invitrogen) and Alexa Fluor 488-conjugated donkey anti-chicken IgG (1:1000, Sigma). Images were then acquired using a confocal microscope (Zeiss, Jena, Germany). The source and identifiers of the reagent used above are listed in Appendix B. 

## 3. Data Analysis

The fluorescence emission was sampled at 100 Hz with the custom-built optical fiber set-up and software, and the behaviors were captured at 30 Hz with two cameras at a spatial resolution of 1280×720 pixels (LRCP10190, China). This divergence of frame rates could lead to a maximal system error of ~33 ms when we calculated the response latency. Both datasets were analyzed by customized software based on the LabVIEW platform (National Instruments, Austin, TX, USA). The optical fiber recordings and the corresponding behavior videos were synchronized using an LED marker visible to the behavior cameras at the start and end of fiber recording. Subsequently, the optical fiber data were low-pass-filtered by a Savitzky-Golay FIR smoothing filter with 50 side points and polynomial order of 3. The Ca^2+^ signals were calculated as relative fluorescence changes: ∆F/F = (F_signal_ − F_baseline_)/F_baseline_, where F_signal_ is the real-time fluorescence intensity and F_baseline_ is calculated as the average of the fluorescence intensity of the entire recording.

For all experiments, the frequency was calculated using peak detection based on a peak detection method described in previous studies [30]. In a multi-spike set, the highest peak is counted as an activity event. If the peak amplitude of the adjacent peak is less than 75% of the highest peak amplitude, it is counted as another. The amplitude of an activity event is calculated as the average amplitude over a 250 ms time window (−120 ms to 130 ms relative to the peak). In awake experiments, the peak of a Ca^2+^ signal was first determined as defined above, and a 10 s time window (−5 s to 5 s relative to the peak) was isolated. Then, the video data were manually screened to check for spontaneous behavior within the time window. The video was then analyzed frame by frame to find the interval of two frames within which the largest displacement of the center of forelimb or head occurred, and the former frame was defined as the behavior onset. For further analysis, a 5 s time window (−1 s to 4 s relative to the behavior onset) was exported. The correlations between Ca^2+^ signals and behavior were analyzed with Igor Pro (WaveMetrics, Inc. V7.0.8.1), MATLAB 2016b (The MathWorks, Inc., Natick, MA, USA), and GraphPad Prism 8. For all tests, *p* < 0.05 was considered to indicate statistical significance. 

## 4. Results 

To establish a protocol for the direct optical measurement of CF activity in freely behaving mice, we used a combination of neuroanatomical techniques, AAV tools, calcium imaging, and optical fiber photometry (Figure 1). Briefly, we first used anterograde tracers (AAV2/8-hSyn-EGFP-WPRE-pA and AAV2/9-hSyn-mCherry-WPRE -pA) to identify and confirm the stereotactic coordinates of the IO subregions that densely projected to lobule IV/V of the cerebellar vermis. Following this, we injected AAV2/9-hSyn-axon-jGCaMP7b-WPRE-pA into the precisely identified IO stereotactic coordinates. After 4 weeks of expression, CF terminals in lobule IV/V were densely labeled and the optical fiber was implanted. Finally, we fixed the optical fiber and allowed the animals to recover. After 24 h, the Ca^2+^ signals of CFs were recorded in freely moving mice in an open field set-up (Figure 1).

### 4.1. Mapping CF Projections to Lobules of the Contralateral Cerebellar Cortex

We first confirmed the stereotaxic coordinates that resulted in CF labeling in lobule IV/V, our target for optical fiber implantation. To do this, we injected AAV2/8-EGFP into the right IO and AAV2/9-mCherry into the left IO. The resulting bilateral tracing verified the contralateral projection pattern of the CFs (Figure 2A–D, Appendix A), and was in agreement with previously published studies [30]. We found that CFs originated from the contralateral IO, travelled through the inferior cerebellar peduncle (Figure 2C) to the cerebellum, and distributed across parasagittal zones within the cerebellum. When we targeted the caudal–lateral regions of the IO, namely the medial accessory olive and caudal regions of the dorsal and principal IO (Figure 2B and Appendix A), we found a high density of resulting CFs labeled in the cerebellar vermis, in agreement with [31,32]. In particular, we found the highest density of resulting CFs labeled in lobule IV/V (Figure 2C, Appendix A), with over a third of the total CF density concentrated within this lobule (35.56% ± 4.80%, Appendix A). Additionally, the density of CF labeling could be consistently and repeatedly produced with targeted stereotaxic injections across mice (Appendix A). We further confirmed these results with injections of a retrograde tracer (CTB 555) into lobule IV/V of the cerebellar vermis (Figure 2D,E). Lastly, we performed immunostaining using PCP4 and Calbindin-D28k staining to reveal the layer of PCs and confirmed that there was no spread of the AAV injection to the cerebellar cortex during injection into the IO, as no PC somata were double-labeled (Figure 3). 

### 4.2. Recording of CF Ca^2+^ Signals in Freely Behaving Mice

The combination of the optical fiber technique and GECI expression allowed us to record CF Ca^2+^ signals simultaneously with active animal behaviors in the open field (Figure 4A). Positioning two cameras in diagonal corners of the open field environment allowed us to capture the behavior of the mice in detail without interfering with their free movement (Figure 4B). Behavior-dependent high-amplitude changes in fluorescence intensity (ΔF/F) were observed in the experimental group (jGCaMP7b, i.e., Ca^2+^ signals) but not in the control group (EGFP, i.e., movement artifacts) (Figure 4D,E, Mann-Whitney U test, *p* < 0.001, *n* = 35 events randomly selected events for each group). As a within-group control that signaled that changes did not result from movement artifacts, we also found that the mean amplitude of fluorescence changes observed from GECI expression (jGCaMP7b, green channel, λ = 470 nm; ΔF/F 1.98% ± 0.01%) was significantly higher than those of red-shifted autofluorescence within the same mice (jGCaMP7b, red channel, λ = 580 nm; ΔF/F 0.72% ± 0.01%; Mann-Whitney U test, ** *p* < 0.05, *n* = 3 mice, averaged across the entire recording session). These behavior-dependent Ca^2+^ signals were also consistently observed over time, across multiple days (Figure 4F). There were no significant differences in the amplitude or frequency of Ca^2+^ signals recorded on day 1, day 2, and day 5 (Figure 4G, ordinary ANOVA test, *p* = 0.9780; and Figure 4H, Brown-Forsythe and Welch ANOVA test, *p* = 0.0825; *n* = 4 mice), indicating that this method can be used to robustly examine CF Ca^2+^ signals during learning paradigms across multiple training and testing days.

### 4.3. CF Ca^2+^ Signals in Lobule IV/V Were Synchronous with Exploratory Motor Behaviors

To examine whether the change in CF Ca^2+^ signals was associated with specific motor behaviors, we classified the behavior of mice in the open field into specific motor motifs, defined as “tiptoe standing,” “rearing,” “grooming,” “turning,” and “locomotion” (Figure 5). We defined the onset of “tiptoe standing” or “rearing” as the moment when the front paws of the mouse left the ground. If the mouse proceeded to stand on their toes, the behavior was defined as “tiptoe standing”; otherwise, the behavior was defined as “rearing” (Figure 5A,B). We then tested the correlation of the change in CF Ca^2+^ signals with each instance of the corresponding behavior (Figure 5C,D). We found that the Ca^2+^ transients were highly correlated with “tiptoe standing” (53 of 61 events) or “rearing” (42 of 45 events), but not with the other observed behaviors (2 of 30 events of grooming, 1 of 34 events of turning, or 3 of 77 events of locomotion) (Figure 5D; Brown-Forsythe and Welch ANOVA test, tiptoe standing vs. grooming, turning, and locomotion, *p* < 0.001; rearing vs. grooming, turning, and locomotion, *p* < 0.001; *n* = 4 mice). In addition, the average peak amplitude of CF Ca^2+^ signals for tiptoe standing or rearing was significantly higher than that of the other observed behaviors (ordinary ANOVA test, tiptoe standing vs. grooming, turning, and locomotion, *p* < 0.001; rearing vs. grooming, turning, and locomotion, *p* < 0.001), with no significant difference between the two identified behavioral motifs (Figure 5E, ordinary ANOVA test, tiptoe standing vs. rearing, *p* = 0.9581). Therefore, these results suggest that the CF activities are likely relevant for exploratory behaviors.

## 5. Discussion

In this study, we established a robust experimental procedure to record the Ca^2+^ signals of CFs, which is a joint method based on neural tracers [33], GECI [34], stereotactic injection [35], optical fiber recording [29,36], and animal behavior [37]. Among these technologies, anatomical tracers verified the key parameters of injection and recording, the axon-specific GECIs reflected the activity signal of CFs, the stereotactic injection ensured the targeted expression of CFs in the cerebellar cortex, the optical fiber photometry ensured the sampling rate and accuracy of the recording, and the behavioral paradigm helped to reveal the biological relevance of the CF signals in freely behaving mice. This approach will be beneficial to directly correlate CF signals to specific behaviors and to provide insight into the fundamental principles of cerebellar processing during behaviors in freely moving mice.

Our method is based on the application of anterograde tracing viruses. Previous studies using traditional tracers and tracing viruses have shown that the majority of the IO projections terminate in the contralateral cerebellar cortex [38,39] and are distributed in a zonal topography [39,40,41]. This is consistent with our results and demonstrates the suitability of the approach for targeting specific cerebellar regions across varying laboratory conditions. Thus, injections can be accurately targeted to specific lobules of interest, e.g., lobule IV/V in the current study, and stereotaxic coordinates then support the application of axon-specific GECIs into specific IO subregions projecting in a topographic manner. For optical fiber recording, it is important to maintain a consistent density of fibers across animals and within the target region of interest; therefore, these topographical tracing methods are useful to establish reliable and reproducible results. Further, our results confirmed that the CFs distributed in the lobules of the cerebellar vermis were dense enough to support the labeling of CF axons with GECIs so that Ca^2+^ signals could be recorded with optic fiber photometry. 

Benefiting from the advantages of optical fiber recording, the highlight of this study is the combination with freely moving behavior. Due to various technical challenges, experiments assessing olivocerebellar activity have largely be conducted in anesthetized animals. While some have suggested that synchrony in spontaneous activity is a robust feature of Purkinje cell complex spike activity in both the awake and anesthetized state [42], others have found that complex spike synchrony patterns specifically change during movement [43,44] and fundamental differences in cerebellar activity have been reported across anesthetized and awake states [45]. This highlights the importance of the current study and the protocol established for an accessible experimental methodology to further investigate olivocerebellar activity in behaving animals. The olivocerebellar circuitry has so far been studied largely using a limited number of behavioral paradigms, such as the forelimb aiming task [46], the footprint pattern [24], and eye blink conditions [47]. Here, we utilized an open field environment as the behavioral paradigm to study CF Ca^2+^ signals that reflect spontaneous independent movement, but our methods can be easily adapted to investigate a diversity of behavioral paradigms. 

The dynamics and correlation of CF activity to various behaviors are also likely to change depending on the specific region or lobule of investigation, as functional specializations have been described across the cerebellar cortex [48]; however, a unified consensus on the functional role of each lobule in behaving animals is far from being obtained [49,50]. The function of lobule IV/V has been traditionally related to limb movement [41,51,52]. In the current study, mice often showed tiptoe standing (body elongation), rearing, grooming, turning, and locomotion in the open field environment. While grooming, turning, and locomotion all require limb movements, the onset of these behaviors were not directly correlated to the CF Ca^2+^ activity using optical fiber recordings. In contrast, larger amplitudes of the CF Ca^2+^ signals, likely resulting from more synchronous CF Ca^2+^ activity, were reliably detected when the mice were rearing or tiptoe standing. According to previous studies, rearing [53,54] and tiptoe standing [55] are considered typical behaviors related to exploration [56]. Therefore, these CF Ca^2+^ signals in lobule IV/V may be related to exploration in the open field—a role that has recently been proposed for this cerebellar region [52]. In future, more specialized behavioral paradigms could be applied using our method in the study of the olivocerebellar circuitry in freely moving conditions.

It should be noted that our method is easy to establish and maintain, in addition to being low-cost, especially compared with two-photon microscopy and other imaging methods, such as miniscope imaging. Moreover, our method could be easily updated with further progress of the applied technologies, or in combination with new techniques. For example, more specific viral markers can be developed along with genetically modified animals, or more specific functional studies conducted in combination with optogenetic or chemogenetic techniques to further probe the functional aspects of the CF neuronal circuitry. Our method could also be easily employed with animal models of cerebellar defects and diseases, such as Cbln1-null mice [57], En2-null mice [58], and cerebellar ataxia mice [59], and allow for the fast screening of interventions or drug treatments in these disease models. In the present study, we focused on recording in lobule IV/V; however, this method can also be easily extended to other cerebellar vermis lobules or even the cerebellar hemispheres in more sophisticated behavioral paradigms, extending the area of study to fields such as social cognition [60,61] and emotion [62,63].

## 6. Conclusions

We have developed an optical fiber-based method to record climbing fiber (CF) Ca^2+^ signals in freely behaving mice, which allows for the robust functional measurement of CF activity from targeted cerebellar regions. Ultimately, we examined how these CF Ca^2+^ signals are synchronized with specific behavioral motifs in an open field paradigm. First, we determined the optimal stereotactic coordinates and adequate density of axonal labeling in lobules IV/V of the cerebellar vermis and then recorded CF Ca^2+^ signals in freely moving mice. We found various movement-evoked CF Ca^2+^ signals, but the onset of exploratory-like behaviors, including rearing and tiptoe standing, was highly synchronous with recorded CF activity. In previous studies, functional measurements of olivocerebellar circuitry were tightly restrained to the use of technically challenging equipment and methodologies, such as electrophysiological recording or two-photon imaging. However, our optical fiber recording method is robust across days, highly accessible, easy to maintain, and low-cost. Our approach provides an accessible method for directly recording the Ca^2+^ activity of a population of CF afferents to a well-defined cerebellar cortical region, which will prove essential for a comprehensive understanding of the functional topography of the cerebellar cortex. Taken together, our approach represents a flexible tool for studying the fundamental cerebellar function and related cerebellar disorders in freely moving animals.

## Figures and Tables

**Figure 1 biology-11-00907-f001:**
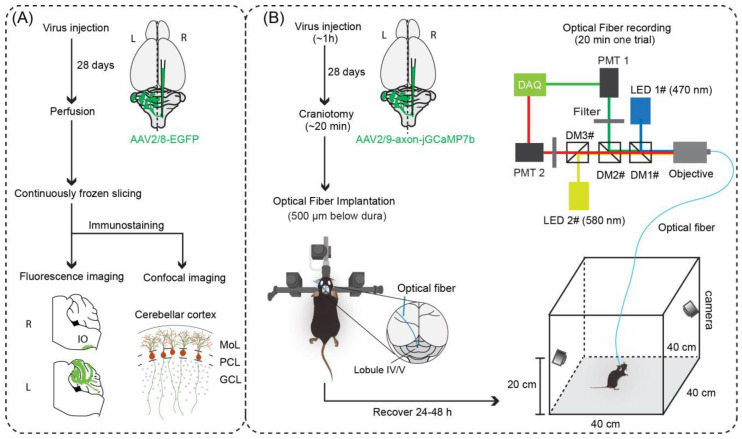
Overview of optical fiber-based recording of climbing fiber (CF) Ca^2+^ signals. (**A**) Experimental procedures to verify the projection and distribution pattern of CFs in the cerebellar cortex. (**B**) Injection of genetically encoded calcium indicators optimized for axonal imaging and optical fiber-based recording of functional CF Ca^2+^ signals in freely behaving mice in an open field environment with behavioral monitoring. R, right cerebellum; L, left cerebellum; LED, light emitting diode (Cree XP-E LED); DM1#, DM2#, and DM3#, dichroic mirror (MD498; Thorlabs); objective (NA = 0.4, Olympus); filter (MF525-39, Thorlabs); PMT, photomultiplier tube (H10721-210, Hamamastsu); DAQ, data acquisition (THINKERTECH). Camera (IR-cut, Tong Xing Yuan, China).

**Figure 2 biology-11-00907-f002:**
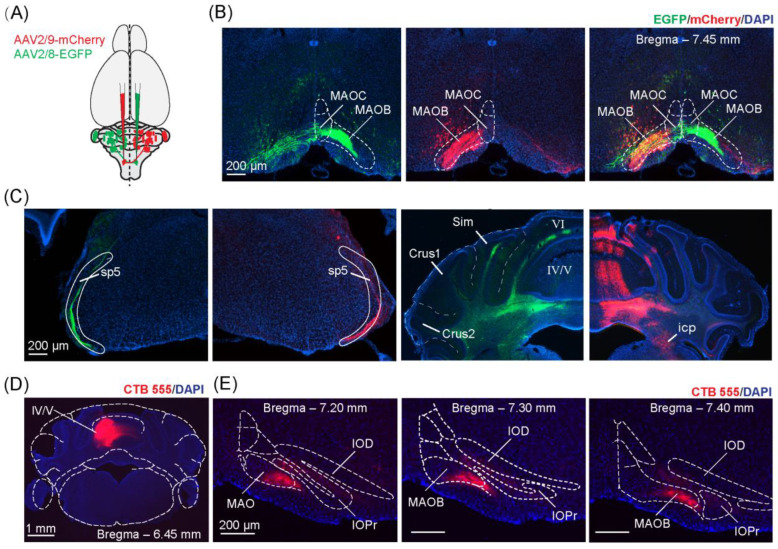
Verification of olivocerebellar projection to specific cerebellar lobules using anterograde tracers. (**A**) Schematic diagram of the bilateral injection of the adeno-associated viruses (AAV). AAV2/8-hSyn-EGFP-WPRE-pA and AAV2/9-hSyn-mCherry-WPRE-pA were used. (**B**) Coronal section showing virus expression in the caudal IO at the injection site. Scale bar = 200 μm. (**C**) Coronal section showing anterograde labeling following injections into the IO depicted in (**B**). Scale bar = 200 μm. sp5, spinal trigeminal tract; icp, inferior cerebellar peduncle; sim, simple lobule; crus1, crus 1 of the ansiform lobule; crus2, crus 2 of the ansiform lobule; VI, lobule VI of the cerebellar vermis; IV/V, lobules IV and V of the cerebellar vermis. (**D**) Coronal section showing the injection site of the retrograde tracer (CTB555), injected into lobule IV/V of the cerebellar vermis. Scale bar = 1 mm. (**E**) A series of coronal sections illustrating retrograde labeling following the injection in lobule IV/V of the cerebellar vermis depicted in (**D**). IOPr, principal nucleus of the inferior olive; IOD, dorsal nucleus of the inferior olive; MAO, medial accessory olive. Scale bar = 200 mm.

**Figure 3 biology-11-00907-f003:**
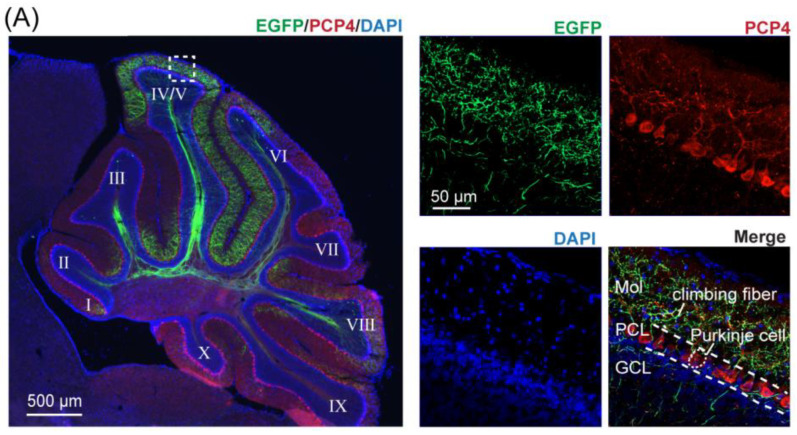
Resulting CF distribution in the cerebellar cortex following injections into the inferior olive (IO). All parasagittal brain sections are from one mouse following an injection of anterograde tracer into the caudal IO. (**A**) Confocal images of the cerebellum immunostained with anti-PCP4 (**A**) or anti-Calbindin-D28K (**B**) shown in red and CF labeling in the molecular layer shown in green following injection of AAV2/8-hSyn-EGFP-WPRE-pA into the IO. Left, scale bar = 500 μm. Right, enlargement of the dotted box in the left image, scale bar = 50 μm. MoL, molecular layer; PCL, Purkinje cell layer; GCL, granule cell layer.

**Figure 4 biology-11-00907-f004:**
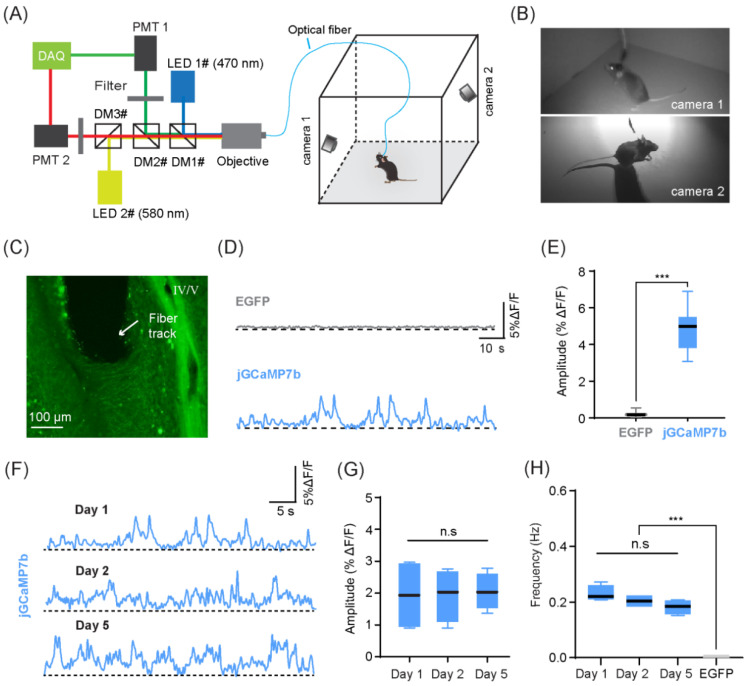
Optical fiber-based Ca^2+^ signal recording of climbing fibers (CFs) in freely behaving mice. (**A**) Schematic of the optical fiber recording and the open field behavioral set-up. (**B**) The actual view of mouse movement captured by two cameras. (**C**) Fluorescence image of a sagittal brain slice of lobule IV/V with CFs labeled with jGCaMP7b. White arrow indicates the position of the optical fiber. Scale bar = 100 μm. (**D**) Example of EGFP and jGCaMP7b fluorescence changes (ΔF/F) in lobule IV/V during the freely moving state. (**E**) Comparison of the amplitudes of EGFP and jGCaMP7b fluorescence changes (Mann-Whitney U test, *** *p* < 0.001, *n* = 35 events for each group). (**F**) Long-term Ca^2+^ recordings over 5 days. Three example traces that were obtained when the mouse was freely moving in an open field on day 1, 2, and 5 after fiber implantation. (**G**) Comparison of the average amplitude of Ca^2+^ signals over 5 days (ordinary ANOVA test, *p* = 0.9780, *n* = 4 mice). (**H**) Comparison of the frequency of Ca^2+^ transients over 5 days (mean number of transient peaks occurring across 25 randomly sampled time windows (5 s each); Brown-Forsythe and Welch ANOVA test, *p* = 0.0825, *n* = 4 mice). EGFP as control (Brown-Forsythe and Welch ANOVA test, *** *p* < 0.001, *n* = 4 mice).

**Figure 5 biology-11-00907-f005:**
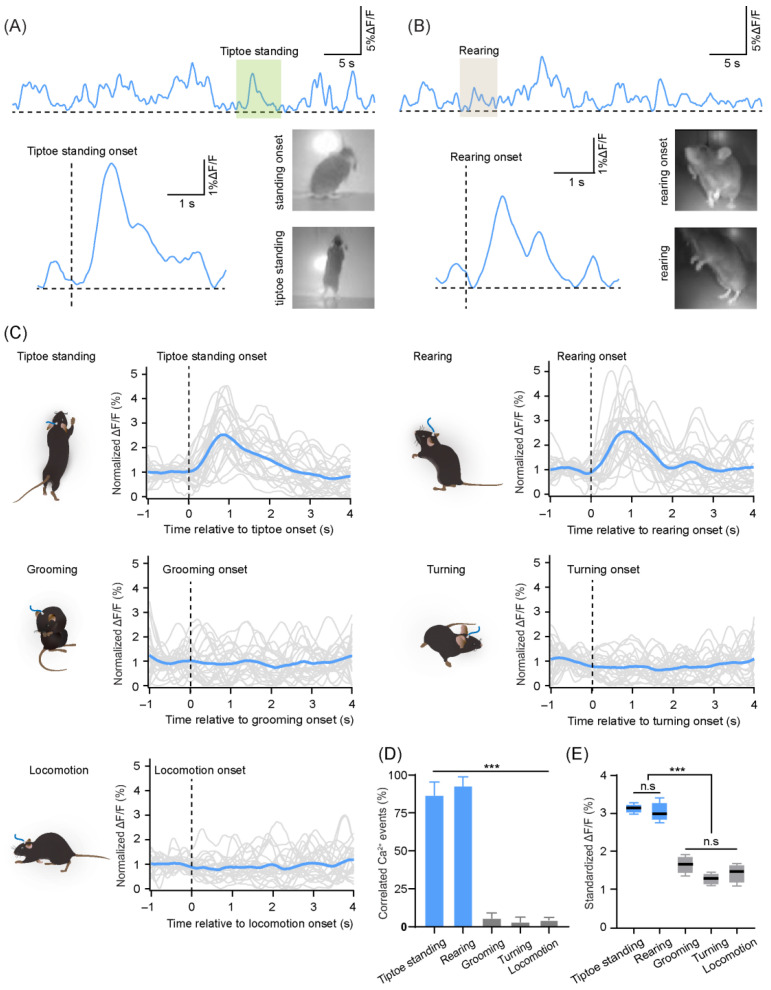
Correlation between various behaviors and the corresponding climbing fiber (CF) Ca^2+^ signals. (**A**) Example behavioral onset and the Ca^2+^ signal of event “tiptoe standing”. (**B**) Example behavioral onset and the Ca^2+^ signal of event “rearing”. (**C**) Five types of behaviors were summarized from 60 min videos recorded in the open field (25 events from four mice for each behavior). Ca^2+^ signals for each event (grey lines) are aligned to behavior onset and the average fluorescence changes (ΔF/F) across all events are shown (blue line). (**D**) The correlation of Ca^2+^ transients with each behavioral motif as defined in panel (**C**). The correlated Ca^2+^ events were detected in a time window of 500 ms after the onset of behavior. There were 53 of 61 events of tiptoe standing (86.7 ± 17.3%), 42 of 45 events of rearing (92.8 ± 12%), 2 of 31 events of grooming (5.7 ± 7%), 1 of 34 events of turning (3.1 ± 6.3%), and 3 of 77 events of locomotion (5.3 ± 3.8%); *n* = 4 mice, 60 min each. Comparison between groups is presented (Brown-Forsythe and Welch ANOVA test, tiptoe standing vs. grooming, turning, and locomotion, *** *p* < 0.001; rearing vs. grooming, turning, and locomotion, *** *p* < 0.001). (**E**) Average amplitude of the peak CF Ca^2+^ signal following the behavior onset in a 2 s time interval for each trial, with averages across all trials for the five behavior types (ordinary ANOVA test, tiptoe standing vs. rearing, *p* = 0.9581; tiptoe standing vs. grooming, turning, and locomotion, *** *p* < 0.001; rearing vs. grooming, turning, and locomotion, *** *p* < 0.001; *n* = 4 mice).

## Data Availability

All data are available from the authors upon reasonable request.

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
