# Peer review of "Optical Fiber-Based Recording of Climbing Fiber Ca2+ Signals in Freely Behaving Mice"

_biology, 2022, doi:10.3390/biology11060907_

Round 1

Reviewer 1 Report

  1. It is unclear regarding the scientific premise that authors would like to present.
  2. The projection maps of the climbing fibers that were demonstrated by the anterograde tracers were not compelling.
  3. Despite the various movement-evoked Ca signals, revealing the neuronal circuits that climbing fibers participate in would strengthen the merit of the manuscript. 

Reviewer 2 Report

General Comments. In their manuscript “Optical fiber-based recording of climbing fiber Ca2+ signals in freely behaving mice” Tang et al., present a procedure with methods based on optical fiber photometry in combination with genetically encoded calcium indicator jGCaMP7b to record chronic activity of climbing fiber (CF) projections to the cerebellar cortex. After establishing the proper stereotaxic coordinates and verifying the organization of contralateral CF projections in fixed brain slices, in vivo Ca2+ signal recordings were performed. Ca2+ transients were recorded in unrestrained isoflurane-anesthetized and additionally in freely moving and behaving C57BL/6 mice. Various movement-involving behaviors were identified and correlated with CF terminal Ca2+ activity.

The procedures are predominantly well presented and conducted, especially the establishment of the proper stereotaxic coordinates and the verification of the organization of contralateral CF projections. Additionally, most of the results and conclusion drawn are clear-cut and the manuscript is well written and presented. The cited literature seems appropriate but could be improved. However, despite my enthusiasm for this manuscript my major concerns are arising due to technical flaws in the acquisition, analysis and interpretation of the data in the anesthetized state. Missing detailed description of the Ca2+   signal detection and analysis routines as well as the comparison and the interpretation of the awake data in relation to the current electrophysiological literature should be added.

In the sections below, several questions and changes or amendments are required to improve the manuscript.

Specific comments:

Major concerns related to the anesthetized recordings:

From the literature of Ca2+ imaging experiments of other brain regions and neurons one would expect higher basal activity and stronger responses to sensory stimuli in the anesthetized animals compared to the awake behaving mice.

When comparing the awake data and the lightest anesthesia of 0.6% isoflurane then the opposite effect is seen with increasing isoflurane concentrations as expected: namely the lightest anesthesia should more resemble the awake situation with sparse transients and increasing amplitudes. (In Figure 5, the frequency of transients as shown in Figure 4, should be implemented, too).

So, the presented data seem not to match the anticipated results. Explanation? Duscussion?  Upon further inspection, it is stated that the heart rate in bpm (Lines 288-289) decrease from 102 bpm at 0.6% iso to 42 bpm at 1.8 % iso (Figure 4C). Given a normal heart rate of an awake mouse of 500-700 it must be assumed that the mice at 42 bpm are near dead. Even the start at 102 bpm is already critical, questioning the physiological state of the animals and the trustworthiness of the acquired data.

Probably, the use of air (Line 117) instead of 100% pure oxygen to deliver the isoflurane was not adequate. Thus, all anesthetized data shown here cannot be used for presentation.

My suggestion would be, to redo the awake and anesthetized measurements in the same implanted mice for at least another 3 mice, for proof of principle, with adequate humidified anesthesia delivery, and temperature control.

The use of animals with an implanted fiber would have the benefit to record the same CF terminals in two conditions for direct comparison and additionally, the total time of anesthesia application can be reduced to the recording time by starting the recordings after recovery from the surgery (fiber implantation). As it has been shown in Figure 5G, the Ca2+ transients  are reproduceable over time (several days at least).

Further requests:

Include additional information in the Material and Method or as Supplementary material section about:

-       C57BL/6 substrain: J or N or ?,

Were animals purchased or generated by in house breeding?

-       Surgeries - Virus injection, Craniotomy and Imaging experiments - durations and anesthesia times (maybe add as a timeline for Figure 1 B & C experiments)

-       Q: For how long the 2.5% isoflurane was given to induce anesthesia?   - Lines 117-118

-    Q: How was the depth of anesthesia monitored over the curse of experiments?

-       Q: How was the body temperature controlled during the anesthetized experiments? (Note: very critical for good physiological condition under anesthesia)

-    Q: Is it true, that the experiments for the anesthetized state was conducted as a terminal experiment, where the animals did not recover from the craniotomy and fiber recording session? => resulting in quite long anesthesia application durations?

-       Q: How was the implanted fiber secured and reconnected to the recording setup for the open field recordings between different days?

-       Q: For the implantation of the optical fiber, was there a similar approach used as described in Li et al., 2019 Long-term Fiber Photometry for Neuroscience Studies, DOI: 10.1007/s12264-019-00379-4? Cite appropriate literature or describe in more detail

-       “The optical fiber was glued to a cannula (inner…”          correct?         - Line158

-       Q: Was the fiber implanted in the anesthetized experiments as stated in the figure legend (Line 285) or “only” advanced to the dura surface (touching it? and/or in immersion with the 0.1M PBS?) as stated in section 2.5  - Lines 159-161

-       Q: How could the isoflurane be set to 0.6% (Line 163) to start the experiments?  Was isoflurane just reduced to 0.6% from the maintaining concentration of 1.0-1.2% (stated in Line 119) used for the surgery and fiber approach? Or how is “initially set to 0.6%” (Line 163) to be understood?

Q: How was the respiratory rate measured/determined to lead to a “(consistent respiratory rate” (Line165)?

Q: How fast were the respiratory rates at each iso concentration until starting the fiber recordings? 

-       Q: Howwere peaks detected? Which Peak detection method/algorithm was used to identify activity events for frequency and amplitude analyses shown in Figure 4C-F?         - Line 220

-      Q: Fbaseline calculation from which region of the recording window?  - Lines 218-219

-        Q: How were the ΔF/F amplitudes determined?

-       Q: Which spiking behavior of the CF fiber terminals do you expect from electrophysiological recordings of previous studies (literature)?  High or low activity under anesthesia in vivo or in brain slices?

-      Q: Does these fit with your recorded activity?  Discuss results in more detail

-       Some statistical comparisons between groups are missing, e.g. in Fig. 4E + 4F between 0.6% and 1.8%  

-       Q: You need to discuss your results in more detail, and integrate them in the current literature about the CF function. Explicitly, for the interpretation of the data, how is the CF terminal activity expected from previous electrophysiological recordings (literature)?  Is the outcome anticipated for the reported anesthetized data? How does the awake, open field data fit into the bigger picture?

The 1% or 5% ΔF/F scale bars of Figures 4, 5 and 6 seem to be extremely small changes in fluorescence intensity upon neuronal activity movement or anesthesia, when taking into account that the jGCaMP7b indicator should report 40% ΔF/F0 for single action potentials and about 200% for 10 action potentials (Dana et al. 2019, DOI: https://doi.org/10.1038/s41592-019-0435-6). Thus, the y-axes of the figures seem to express fold change compared to baseline (resting Ca2+ levels) while the labels stating %.

To the results in Figure 6: According to the legend (Lines 347-348) “average fluorescence changes (ΔF/F) across all events are shown (blue line in Fig 6C) while in Figure 6D “Summary of the average amplitude of CF Ca2+ signals are depicted. Upon further inspection, the Mean amplitudes in 6D does not match the blue line amplitude values (peak minus baseline?) of 6C.  Why?

Q: How are the amplitudes calculated?

Q: What does normalized ΔF/F (%) stand for?    Normalized against what?

Minor concerns:

Spell/Fond correction:

- Line 141       three times

- Line 143       Metacam; Germany)

- Line 173       (PALA; Germany).

- Implement info about MATLAB version and toolboxes used - Line 227

- Lines 259-260 in Figure 3 Legend – “All brain sections are from one mouse” which was injected with xy “into the IO”     -   sentence somehow incomplete

Reviewer 3 Report

In this study, titled “Optical fiber-based recording of climbing fiber Ca2+ signals in freely behaving mice”, Tang and Xue et al. propose an optical fiber-based method to record neuronal activity from cerebellar climbing fibers, which allows functional measurement of CF in freely behaving mice. The work is quite well written and describe with sufficient detail the methodology employed, however I have some major points I would like the authors to address before endorsing it for publication in Biology.

Major comments:

  • Reference and presentation (lines 78-95) of Figure 1 should be moved from the Introduction to the Results section, leaving in the Introduction only a general description of the work done.
  • Section 2.1: please report the comprehensive number of animals used in the study.
  • Section 2.4: since the authors report the use of a custom-built optical fiber recording setup, they should thoroughly describe the optical system in all its components, at least all of those shown in Figures 1C and 5A.
  • Section 2.6: please provide model and brand of the camera employed, and of the objectives if any were employed.
  • Section 4.3/4.4: I think it would be beneficial to the work to insert at least a video of an animal exploring the open field environment with the optical fiber mounted. Exemplary videos of the mentioned behaviors (e.g., tiptoe-standing, rearing, etc.) would be welcome too.

Minor comments:

  • Figures 1C and 5A, correct “fifter” with filter.
  • Line 134: “using by a glass micropipette”, remove “by”.
  • Line 234: “AAV2/9-mCherry was into the left IO”, remove “was”.
  • Discussion, lines 358-362: please check figures numbers reported, as there is no Figure 7 in the manuscript.
  • Mention to the content of Appendix A should be made in the text.

Reviewer 4 Report

The topic of this paper is relevant, timely, and of interest to the audience of this journal. The introduction to this manuscript provides a comprehensive overview of the recent advances in the subject area and dwelves into some of the technicalities and methods in studying the workings of the olivo-cerebellar circuitry. The graphical abstract to the proposed protocol is neatly laid down and summarized in a four-stage process that include (i) neural tracer labelling of the cerebellar cortex (ii) the attachment of the optic fiber (iii) the expression profile of the GECIs and (iv) the correlation of the calcium signals to the motor coordination and behaviour patterns in mice.

The manuscript is technically correct and the methods are used correctly so that it is in my opinion that the data is sufficient to corroborate the claims. The reporting appears to also be  sufficiently transparent to repeat the experiments for those not familiar with these types of experiments. In my opinion, this work should make an impact on the research field covered in this exciting emerging area of science as it does away with the use of technically challenging equipment and methodologies that can resolve intrinsic calcium signals with behavioral patterns.  Such a study could also be tied to neurovascular coupling in disease states and cerebellar plasticity through motor learning. One important feature that seems to be missing from this study and which would attract a greater readership would be the inclusion of a short video showing the tandem in vivo sampling of the Cacium transients with the various correlated behaviours and the setup in general.

Round 2

Reviewer 1 Report

1. The authors should reconsider to submit their manuscript to the journals covering the technologies related to biology.

2. Manuscript editing service may be required to facilitate the publication.

Author Response

Point 1. The authors should reconsider to submit their manuscript to the journals covering the technologies related to biology.

Response 1: We strongly believe that our manuscript is suitable for publication in Biology. Indeed, the stated scope of the journal covers the technologies related to biology:

“As well as standard research articles we will also publish reviews, commentaries, hypotheses and descriptions of new methods.

Point 2. Manuscript editing service may be required to facilitate the publication.

Response 2: We have carefully and thoroughly gone over the manuscript, correcting the remaining typos. The manuscript has also been edited by a native English speaker, the co-author Dr. Janelle Pakan.

Reviewer 2 Report

Suggestion after Revision (V2):

In general, the thoroughly revised version (V2) has much improved the current manuscript and most of the raised concerns were adequately answered and implemented. However, some aspects remain unaddressed or unsatisfactory.

To Point 3:

Being aware, that the comparison of the measured Ca2+ transients in the anesthetized and the awake (behaving) state was not the major topic of the paper (as different fiber placements and spontaneous vs movement-initiated (no movement artefacts) signals were chosen to be acquired), the physiological conditions during data acquisition, however are strongly relevant. In this respect, at least the bottom trace in Figure 4C, for 1.8 % isoflurane (42 bpm), even for breathing rate (instead of heart beat) a value of 42 bpm is still very low and critical. Even breathing rates below 100 bpm should be avoided in future experiments.

Maybe this critical physiological condition (very low bpm) led to the surprisingly low frequency and amplitude values reported in Figure 4 E and F, respectively. (This statistically significant (!) drop in monitored Ca2+-transient parameters was just shown but neither mentioned nor discussed).

This should be caught up on.

Possibly, in the course of the already discussed and cited literature with the increasing synchronicity of neuronal activity upon deeper anesthesia, one would expect for 1.8 % isoflurane an even increased amplitude with a slight decrease in frequency which probably might have been statistically significant compared to 0.6 % isoflurane. However, upon a too strong anesthesia and impaired breathing rate, the physiological condition and the Ca2+-transients in turn, dropped dramatically in amplitude and frequency. 

To strengthen the presented results of the awake (behaving) mouse (Figure5 and 6), a frequency graph, similar to Figure 4E should easily be generated, since the amplitudes (and the appropriate peak onsets) were already analyzed and reported. The recorded frequencies of the Ca2+-transients and the actual detected behavioral occurrences of the identified behaviors (tiptoe standing and rearing), should be compared and displayed, too in Figure 5 or 6 (frequency Hz vs EGFP, jGCaMP7b, behaviors), to gain even more insight in the awake, behaving state.                         

Typo:

Line 95:  Figure 1A

Perfusion

Confocal imaging

Figure 1B

Under dura  500µm  below dura

under anesthesia

Figure1C

once one trial

Line 100: - add MD numbers (Thorlabs) for MD #1 and #3  in legend

Line 238:              excess space      peak).    .  In awake

Line 278:              E,)..

Line 423:              (Figure 5 and 6)

Line 427:              imaging [59]

Line 431:              animals [60]

Line 235:              state [60]

Line 236:              movement [62-63]

Line 248:              cortex [65]

Line 249:              understood [66-67]

Line 494:              excess space  fibers    for

Reviewer 3 Report

The authors have addressed all of my comments.

Author Response

We thank the reviewer for the time.